# Residual Stream Analysis of Overfitting And Structural Disruptions

**Quan Liu**[*]
BUPT
qaunliu@bupt.edu.cn

**Han Zhou**
Baidu
hanzhou@baidu.com

**Wenquan Wu**
Baidu
wenquanwu@baidu.com

**Hua Wu**
Baidu
huawu@baidu.com

**Sen Su**
BUPT
sensu@bupt.edu.cn

## Abstract

Ensuring that large language models (LLMs) remain both helpful and harmless poses a significant challenge: fine-tuning on repetitive safety datasets—where unsafe prompts are paired with standard refusal templates—often leads to *false refusals*, in which benign queries are declined. We first quantify this effect, showing that safety data exhibits substantially lower token entropy ($H_1 \approx 9.18$) and 2-gram diversity ($\approx 0.048$) compared to general instruction data ($H_1 \approx 12.05$, 2-gram$\approx 0.205$). To uncover the root cause, we introduce *FlowLens*, a stable PCA-based tool for residual-stream geometry analysis, and reveal that higher proportions of safety examples concentrate variance along a few components, reducing representational smoothness and driving false refusals (false refusal rate rises from 63% to 84% as safety data increases from 0% to 40%). Guided by these insights, we propose *Variance Concentration Loss* (VCL), an auxiliary regularizer that penalizes excessive variance concentration in mid-layer residuals. Empirical results demonstrate that VCL reduces false refusals by over 35 percentage points while maintaining or improving performance on general benchmarks such as MMLU and GSM8K.

## 1 Introduction

Large language models (LLMs) such as GPT-3 [6], PaLM [8], and LLaMA [30] have demonstrated human-level performance across a wide array of NLP tasks, including question answering, summarization, dialogue, and code generation. However, the widespread adoption of these models gives rise to significant concerns about unintended harmful outputs—including hate speech, misinformation, and the facilitation of illicit activities—that can undermine user trust and introduce tangible risks. [12, 4].

To mitigate such risks, a common defense is *safety fine-tuning*: supplementing pre-trained LLMs with curated safety datasets that pair unsafe or adversarial prompts with refusal or safe-completion responses [5, 18, 14]. While safety fine-tuning dramatically reduces overtly harmful generations—blocking over 95% of unsafe prompts on benchmarks like WildGuardTest and Jailbreak-Trigger—it also introduces a new failure mode: *false refusal*, where the model erroneously declines benign queries. On an exaggerated safety prompt sampled from XSTEST (see Figure 1), we observe that Llama-3.2-1B-Instruct produces a refusal even for a benign request. Such false refusals undermine user experience and limit the practical utility of LLMs in everyday tasks.

---

[*]Work done at Baidu during an internship.

39th Conference on Neural Information Processing Systems (NeurIPS 2025).

Figure 1: Examples of false refusal on an exaggerated safety prompt sampled from XSTEST. Our method avoids false refusal and gives an appropriate response. Model and dataset details are provided in Section 5.2.

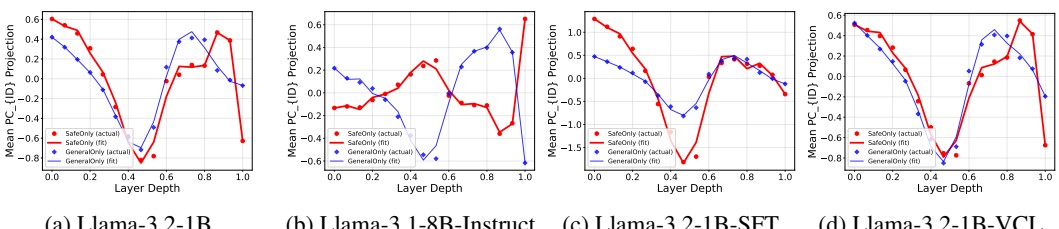

| (a) Llama-3.2-1B | (b) Llama-3.1-8B-Instruct | (c) Llama-3.2-1B-SFT | (d) Llama-3.2-1B-VCL |

Figure 2: Residual trajectories of the mean alignment score along the top principal component using FlowLens for four instruction-tuned LLMs on TruthfulQA (General, blue) versus XSTest (Safe, red). Panels (a)–(d) correspond to: (a) Llama-3.2-1B, (b) Llama-3.1-8B-Instruct, (c) Llama-3.2-1B-SFT, and (d) Llama-3.2-1B-VCL (ours). Each curve plots the projection of the final token's residual vector at normalized layer depth $[0, 1]$. Examples shown above illustrates that standard safety fine-tuning collapses mid-layer variance around depths 0.4–0.6, leading to more false refusals on XSTest; in contrast, VCL stabilizes variance across layers and maintains safety.

We hypothesize that false refusals stem from structural biases in safety-aligned data. In particular, refusal completions are often highly repetitive and templated: across three standard safety corpora (WILDJAILBREAK, WILDGUARDMIX, TULU-3-SFT-MIXTURE), the average unigram entropy is only $H_1 \approx 9.2$, and distinct 2-gram rate is 4.8%, versus $H_1 \approx 12.1$ and 20.5% for general instruction data [17]. Crucially, if we isolate just the completions (excluding prompts), these diversity metrics drop even further. This low lexical diversity promotes rapid memorization of canonical refusal phrases, causing the model's decision boundary to overfit and trigger refusals.

To assess how these biases impact model internals, we introduce *FlowLens*, a PCA-based tool that concatenates residual vectors from a selected window of transformer layers and performs unlayered principal component analysis. When applied to models fine-tuned with varying proportions (0–50%) of safety data, FlowLens reveals a pronounced *geometric collapse*: as the safety ratio increases, variance becomes increasingly concentrated in the top principal component, and the *alignment score* along this axis falls from 0.99 to 0.83 (Figure 5). This collapse, illustrated in detail in Figure 2, correlates strongly with rising false refusal rates, exposing a representational signature of over-caution. Guided by these insights, we propose the *Variance Concentration Loss* (VCL), an auxiliary regularizer that penalizes excessive variance concentration in mid-layer residuals during SFT. VCL preserves the defensive strength of safety tuning by correctly rejecting 98% of unsafe prompts, reduces false refusal rates on XSTest by 35 percentage points, and decreases compliance-refusal errors on JailbreakTrigger by 28%. Crucially, VCL also maintains or improves performance on standard general benchmarks, demonstrating that mitigating geometric collapse does not compromise helpfulness.

**Contributions.**

- We identify and quantify key structural biases in safety-aligned data—low token entropy and n-gram diversity—that drive false refusals.

- We develop *FlowLens*, a stable, unlayered PCA-based tool for residual-stream geometry analysis, revealing how safety data disrupts internal representations.
- We introduce *Variance Concentration Loss* (VCL), a novel auxiliary regularizer for mid-layer residuals, and empirically show its efficacy in substantially reducing false refusal rates without harming general capabilities.

## 2   Related Work

**False Refusal Mitigation Methods.**   Existing methods for mitigating false refusal can be broadly grouped into two categories: *sample-based* approaches and *inference-time adaptation*. Sample-based methods require additional curated data or synthetic examples to fine-tune or calibrate the model, which introduces extra data collection and training costs [28, 7, 34]. Inference-time adaptation methods modify the decoding process or inject runtime interventions to steer model outputs, but they may suffer from distribution shift between training and inference, leading to unstable behavior [39, 37]. In contrast, our approach introduces an auxiliary loss during training, which reduces false refusal without requiring additional training samples or modifying the inference process.

**Residual Stream.**   Prior work has examined the residual stream in the context of safety alignment and in broader geometric analyses. In safety-related studies, researchers have compared the residual representations of safety prompts and general prompts, often focusing on directional differences or cosine similarity between the two [38, 3, 34]. However, such analyses typically overlook the underlying structure of the residual space, leading to instability and inconsistent findings (see Section 4.3).Separately, a line of research investigates the geometry of the residual stream in general-purpose models [27, 22, 32]. These studies often analyze residuals on a per-layer basis, or concatenate residuals across layers into a higher-dimensional trajectory. Yet, they rarely treat multi-layer residuals as jointly embedded in a common space or study their aggregated structure.

## 3   How Structural Repetitiveness in Safety Data Leads to Overfitting

To investigate how the tension between helpfulness and harmlessness manifests in the internal representations of language models, we begin with an analysis of the safety-aligned training data. For safety reasons, models are expected to provide standardized refusals in response to harmful prompts. These refusals often follow canonical patterns such as rejections, disclaimers, or ethical caveats. The consistency of these patterns is reflected in recent jailbreak benchmarks [41, 21, 24] that rely on string-matching against a fixed set of refusal phrases to determine whether a model is aligned.

While some recent work has attempted to improve the diversity of completions through response filtering [25, 2], these efforts are based on heuristic filtering strategies applied after data collection. Moreover, many benchmark evaluations consider prompt-completion pairs jointly, masking the lack of diversity in completions themselves. Since cross-entropy loss during fine-tuning is computed only over the target completion tokens, we argue that it is critical to analyze completion repetitiveness in isolation.

To quantify this structural repetitiveness, we compute a suite of lexical diversity metrics—token entropy, mean segmental TTR (MSTTR), and unique $n$-gram coverage. We follow the methodology proposed in [16]. The lexical diversity metrics used in this analysis are detailed in Appendix A. We use three datasets in this study: WILDJAILBREAK, WILDGUARDMIX, and TULU-MIX, each containing approximately 100,000 safety-aligned completions.We additionally sample 100,000 non-safety examples from the TULU-3-SFT-MIXTURE-GENERAL dataset as a control group. Appendix B provides additional information about each dataset used in our study, including how completions are constructed, filtered, and organized. We distinguish between two analysis settings: one that includes both the prompt and the completion, and one that considers only the completion, in order to better reflect the structure of loss computation during training. As shown in Table 1, safety completions consistently score lower than general completions across all metrics. A full list of the top-25 most frequent trigrams in each subset is provided in Appendix C. These statistics reflect a constrained lexical range and heavy reuse of high-frequency refusal phrases such as "I'm sorry, but..." This linguistic homogeneity narrows the training signal and limits the expressive capacity of the model during fine-tuning.

| Metric | WILDJAILBREAK | | WILDGUARDMIX | | TULU-3-SFT-MIXTURE | | Control | |
|---|---|---|---|---|---|---|---|---|
| | w/o query | w/ query | w/o query | w/ query | w/o query | w/ query | w/o query | w/ query |
| Entropy $H_1 \uparrow$ | 9.18 | 9.41 | 11.11 | 12.30 | 10.05 | 11.22 | 12.05 | 12.18 |
| Entropy $H_2 \uparrow$ | 12.63 | 14.89 | 15.97 | 16.15 | 14.27 | 15.39 | 17.02 | 17.25 |
| Entropy $H_3 \uparrow$ | 13.52 | 15.68 | 15.23 | 16.37 | 14.92 | 15.04 | 18.28 | 18.43 |
| MSTTR$\uparrow$ | 0.672 | 0.689 | 0.637 | 0.645 | 0.659 | 0.674 | 0.753 | 0.767 |
| Distinct 2-gram$\uparrow$ | 0.048 | 0.066 | 0.152 | 0.177 | 0.103 | 0.218 | 0.205 | 0.338 |
| Distinct 3-gram$\uparrow$ | 0.408 | 0.553 | 0.541 | 0.593 | 0.312 | 0.539 | 0.716 | 0.759 |

Table 1: Lexical diversity metrics (entropy, MSTTR, and distinct $n$-gram rates) for each dataset, comparing cases without and with query context. To avoid interference from the dialogue template "User: … Assistant: …" in the with-query setting we count the query and completion as two separate samples.

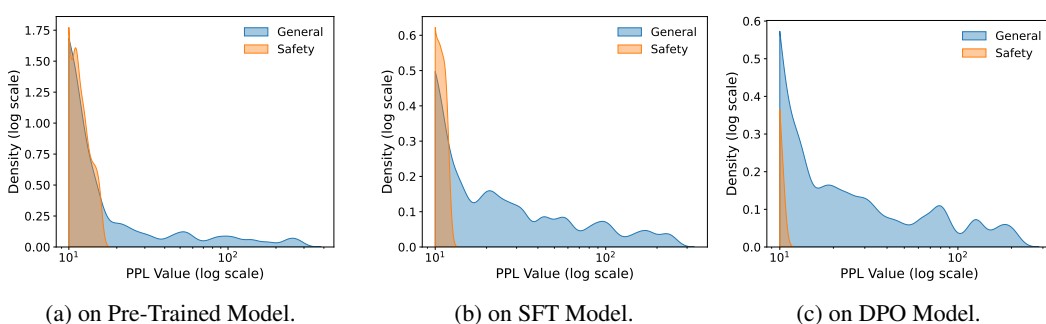

(a) on Pre-Trained Model.    (b) on SFT Model.    (c) on DPO Model.

Figure 3: Loss behavior differences between safety and general tasks. Safety data shows lower average PPL but greater variance and heavier tail. Our experiments employ the Llama-3.1-Tulu-3-8B model family.

We further examine how these low-diversity completions affect the training dynamics of language models. Specifically, we use perplexity (PPL) as a proxy for model confidence. We compute PPL separately over the completions in each example, using models at various stages of alignment. As shown in Figure 3, safety completions consistently exhibit lower average PPL than general completions. However, this is not evidence of easier generalization. Rather, it reflects overconfidence on memorized refusal templates.

More concerningly, we observe that models fine-tuned on repetitive safety data are prone to *false refusals*—they mistakenly reject benign queries with overly cautious completions. This phenomenon is further supported by the instability of principal components shown in Table 2. We interpret this as a form of *structural overfitting*, arising not from insufficient data volume, but from a mismatch between prompt diversity and completion homogeneity.

Overall, our findings reveal a structural mismatch introduced during safety fine-tuning: models are trained on diverse and adversarial prompts, yet learn to produce narrowly templated completions. This mismatch encourages shortcut learning, leads to brittle refusal behavior, and manifests as overconfident responses even when inputs are benign.

## 4  Residual Stream Geometry and Safety Representations

Transformer-based language models communicate intermediate computations through a structure known as the *residual stream* [31, 9]. At each layer, the residual vector carries forward accumulated semantic and syntactic information, making it a rich object for representation-level analysis.

Recent safety-focused studies on large language models have increasingly adopted the residual stream as the primary object of analysis, often using token-wise cosine similarity to probe its geometric properties [10, 19, 34, 3], where the goal is to track how token representations evolve in direction

across layers. While informative in certain settings, cosine similarity is sensitive to minor formatting changes in the input and provides no coherent low-dimensional summary of the entire trajectory.

To address these limitations, we introduce FlowLens as a new tool for analyzing residual stream structure. Rather than inspecting each layer independently, we concatenate residuals from all layers of a prompt into a single high-dimensional vector and perform PCA over the resulting dataset. This approach captures long-range geometric trends, allowing for prompt-wise comparison in a shared coordinate space.

## 4.1 Formalization of Residual Trajectory Projections

Let each prompt $x_i$ produce a sequence of residual vectors $(r_i^{(1)}, \ldots, r_i^{(L)})$ from $L$ transformer layers (we follow prior work [33] and extract the residual vector corresponding to the final token of each prompt), with each $r_i^{(l)} \in \mathbb{R}^d$. We collect residual vectors from $N$ prompts and $L$ layers into a single matrix $X \in \mathbb{R}^{(N \cdot L) \times d}$, where each row corresponds to a residual vector from a particular layer and prompt.[2] Transformer residuals evolve through linear transformations and additive updates across layers [9]. This intrinsic linearity makes PCA a natural analytical choice: it preserves the intrinsic linear geometry of the representation space while extracting its dominant modes of variation [15]. We first center the matrix $X$ by subtracting the mean residual across all rows. We then perform PCA on $X$ to extract the top principal directions $\{\mathbf{v}_j\}$ of its covariance matrix. We refer to this approach as **FlowLens**.

To determine the number of principal components to retain, we estimate the *intrinsic dimension* (ID) of the full residual stream matrix $X$ using the TwoNN method [11]. This approach infers a lower bound on the manifold dimension by comparing ratios of first and second nearest-neighbor distances in the high-dimensional data. Since the computed ID of 2.98 represents the minimal embedding dimensionality, we conservatively round up to 3 when selecting our PCA dimension.

**Experimental Setup.** We evaluate three instruction-tuned language models spanning multiple architectures and scales: LLaMA-3.2-1B-Instruct [13], LLaMA-3.1-8B-Instruct [13], LLaMA-2-7B-chat-hf [30]. As evaluation data, we use the TruthfulQA [20], a widely non-safety adopted dataset. Full statistics and trends on more models are provided in Appendix F.

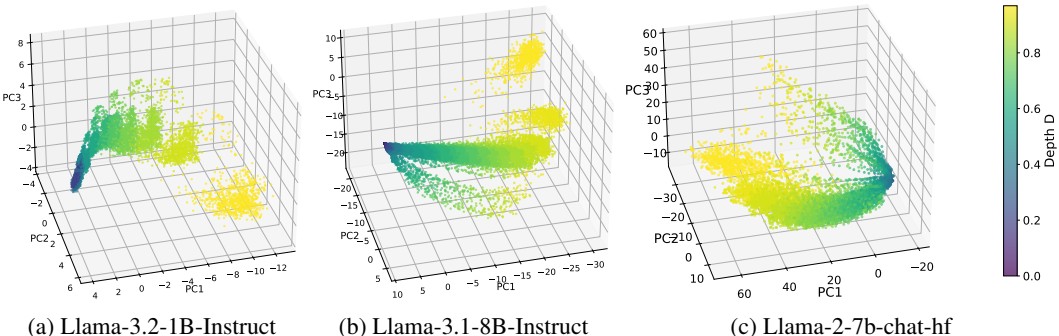

(a) Llama-3.2-1B-Instruct     (b) Llama-3.1-8B-Instruct     (c) Llama-2-7b-chat-hf

Figure 4: Projections of residual trajectories using FlowLens for three instruction-tuned language models on the TruthfulQA dataset. Each point represents the PCA-projected residual vector of the final token from one prompt, colored by its corresponding layer index (depth normalized to $[0, 1]$).

Figure 4 shows the resulting PCA projection of residual trajectories. We observe a **consistent** unfolding pattern across all tested models, each of which adopts a transformer decoder-only architecture. Under FlowLens, the residual stream trajectories form smooth and coherent curves in the PCA-reduced space, with points ordered by layer depth. Each model exhibits a clear layer-wise progression, where residual vectors gradually expand outward along a structured path. Moreover, per-layer residuals cluster in distinguishable zones that grow with depth, reflecting a consistent

---

[2]To avoid spurious effects, we preprocess inputs by removing trailing punctuation (e.g., question marks, periods) before extracting residuals. In this section, all analyses use raw prompt inputs without any chat templates to prevent template-induced artifacts.

representational evolution. Residual trajectories from different models may differ by a global rotation in the PCA space. In transformer architectures, such rotations do not affect the semantics of internal representations, as the residual stream does not possess a privileged basis [9].

To our knowledge, this is the first method to reveal such a **layer-aligned geometric trajectory** in the residual stream. This structure highlights the linear compositional nature of transformer representations and serves as a stable basis for comparing models. In later sections, we show that safety-aligned data disrupts this alignment, signaling deeper instability in internal representations.

## 4.2 Structural Disruptions Induced by Safety Data

To isolate the structural effects of safety-aligned data on the residual stream, we conduct two sets of experiments using FlowLens. In the first setting, we examine a model that has been instruction-finetuned on mixed data, and compare how different subsets of data (e.g., safe vs. general) affect the layerwise evolution of $PC_{ID}$. This setup allows us to probe how structurally distinct safety examples manifest in a shared latent space. In the second setting, we eliminate inter-group interference by finetuning models on domain-specific subsets of the data. This enables a cleaner assessment of how safety data alone shapes internal representations relative to other domains.

To quantify the extent of disruption, we define the structural alignment score as the cosine similarity between the $ID$-th principal component of each domain-specific model and that of a global PCA basis:

$$\cos\theta = \left| \left\langle \mathbf{v}_{ID}^{(\text{model})}, \mathbf{v}_{ID}^{(\text{global})} \right\rangle \right|,$$

where $\mathbf{v}_{ID}^{(\text{model})}$ and $\mathbf{v}_{ID}^{(\text{global})}$ are the unit-norm $ID$-th principal directions from the model-specific and global PCA spaces, respectively. Lower values of $\cos\theta$ indicate greater misalignment and thus a higher degree of structural disruption in the residual space. Note that the cosine is computed between principal component directions rather than directly between residual vectors, as in prior work.

**Experimental Setup.** For both experiments, we use LLaMA-3.2-1B as the base model and LLaMA-3.2-1B-Instruct as the finetuned model. The training corpus is drawn from the Tülu 3 dataset [17], and we follow the open-source Tülu 3 instruction tuning recipe.[3] Each SFT experiment is conducted using 100,000 examples sampled from the corresponding domain subset.

Figure 5 presents the results. In the first row, we plot the $PC_{ID}$ center trajectories for safe and general samples within the same finetuned model. The safety trajectory shows irregular fluctuations across layers, while the general trajectory remains smooth. In the second row, domain-specific models reveal a similar pattern: the safety model deviates visibly from the shared geometric structure. This divergence is supported quantitatively: the $PC_{ID}$ direction of the safety-only model has a cosine similarity of 0.84 with the global basis, compared to over 0.98 for general-aligned models.

To understand how the extent of safety data contributes to instability, we analyze models trained with increasing proportions of safety examples. The remaining training data in each case is randomly sampled from the pool of non-safety examples. For each model, we measure the variance along $PC_{ID}$ and the false refusal rate on benign prompts. False refusals are evaluated on the XSTest benchmark, following the evaluation protocol and decontamination procedure used in the Tülu 3 recipe. The results are summarized in Table 2. As the safety data ratio increases, $PC_{ID}$ variance grows steadily, suggesting increasing distortion in residual geometry. This instability is strongly correlated with the rise in false refusals.

| Safety Ratio | 0.0 | 0.1 | 0.2 | 0.3 | 0.4 | 0.5 | 0.6 | 0.7 | 0.8 | 0.9 | 1.0 |
|---|---|---|---|---|---|---|---|---|---|---|---|
| **Score** | 1.0 | 0.94 | 0.91 | 0.89 | 0.85 | 0.82 | 0.84 | 0.83 | 0.77 | 0.76 | 0.75 |
| **False Refusal** (%) | 0.63 | 0.74 | 0.79 | 0.81 | 0.84 | 0.86 | 0.82 | 0.81 | 0.85 | 0.92 | 0.97 |

Table 2: Effect of safety data ratio on residual structure and refusal metrics. "Score" measures alignment along $PC_{ID}$ using global directions $\mathbf{v}_{ID}^{(\text{global})}$ from the model at safety ratio 0. False Refusal is the precision of rejecting benign prompts.

---

[3] https://github.com/allenai/open-instruct.git

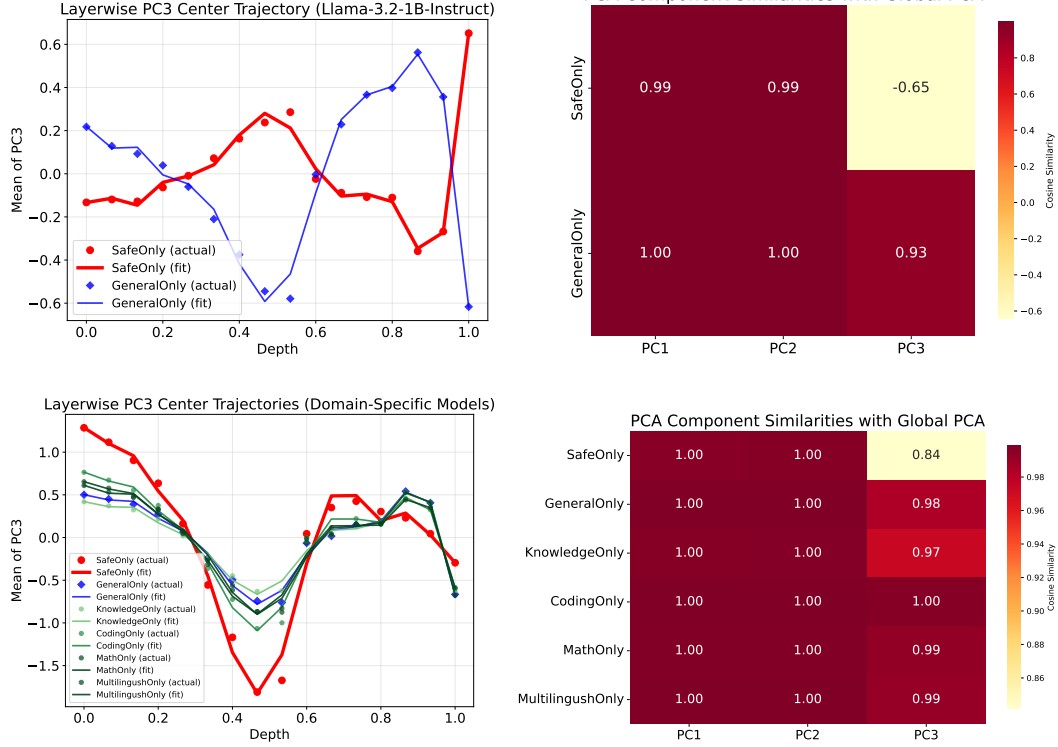

Figure 5: Layerwise $\text{PC}_{ID}$ center trajectories under FlowLens. Top: safety vs. general prompts within the same instruction-tuned model (LLaMA-3.2-1B-Instruct); Bottom: models trained on domain-specific subsets from the Tülu 3 dataset [17]. Safety data produces irregular $\text{PC}_{ID}$ curves, deviating from the smooth, aligned progression seen in general and other domains. These deviations signal a breakdown in residual stream structure caused by safety fine-tuning.

### 4.3   Stability of FlowLens

We propose FlowLens as a stable method for analyzing internal representations in large language models. Unlike cosine similarity, which is highly sensitive to surface-level variations in prompt formatting, FlowLens applies principal component analysis (PCA) to the full residual stream trajectory, capturing the global structure of residual space.

We define *stability* as the consistency of an analysis tool's output under small perturbations of the input prompt that do not alter its semantics. A stable method should yield similar representations or structural patterns—such as principal directions or distances—regardless of minor changes in punctuation, phrasing, or tokenization boundaries.

**Theoretical Justification.**   Let $X \in \mathbb{R}^{N \times dL}$ be the matrix of residual trajectories from $N$ prompts, where each row concatenates the residuals from $L$ layers, each of dimension $d$. PCA computes the top eigenvectors $\{\mathbf{v}_j\}$ of the covariance matrix $\Sigma = \frac{1}{N}(X - \bar{X})^\top (X - \bar{X})$.

For two datasets $X$ and $X'$, representing prompt variants differing only in punctuation, if $\|\Sigma - \Sigma'\|$ is small, then by perturbation theory (e.g., Weyl's theorem [35]), the leading eigenvectors $\{\mathbf{v}_j\}$ will also be close. This implies that projections onto principal components, especially PC1, remain consistent:

$$|\langle \mathbf{x}_i, \mathbf{v}_1 \rangle - \langle \mathbf{x}'_i, \mathbf{v}'_1 \rangle| \ll 1$$

Thus, PCA offers a stable basis for comparing the structure of residuals across prompt variants, while cosine similarity—being a local angle-based metric—is more susceptible to variation from minor surface changes.

**Empirical Validation.**   We validate this stability property using 450 prompts from XSTest [26], where each prompt appears in two forms: one ending with a question mark and one without. Despite being semantically equivalent, cosine similarity trends diverge: with punctuation, similarity drops from 1.0 to 0.6; without, it increases from 0.0 to 0.6 (Appendix E).

By contrast, FlowLens produces consistent PC1 trajectories across both groups. This confirms that PCA projections are insensitive to superficial formatting, and are suitable for analyzing residual geometry in a stable and interpretable manner. Specifically, the PC1 projection correlation between the punctuation and no-punctuation groups exceeds 0.98 across all layers, highlighting the method's stability.

## 5   Variance Concentration Loss

In this section, we propose an auxiliary loss aimed at encouraging structural consistency in the residual stream throughout supervised fine-tuning (SFT). Our design is motivated by the observation that fine-tuning on safety-critical data often leads to structural distortions in the model's internal representation, manifesting as unstable principal directions in the residual space.

Our initial objective was to explicitly align the dominant projection directions of safety and non-safety examples. Given residual matrices $R^{(\text{safe})}$ and $R^{(\text{gen})}$ from the same layer but different data categories, we considered minimizing the distance between their projected subspaces:

$$\mathcal{L}_{\text{align}}^{(l)} = \left\| V_k^{(\text{safe})} V_k^{(\text{gen})\top} - I_k \right\|_F^2$$

where $V_k^{(\text{safe})}, V_k^{(\text{gen})} \in \mathbb{R}^{k \times d}$ are the top-$k$ principal components of the centered residuals from each data type, and $I_k$ is the identity matrix. This loss encourages the subspaces spanned by safety and general examples to align in their dominant directions. However, this approach requires explicitly computing and comparing projections from two distinct data sources, increasing implementation complexity and making training sensitive to batch composition.

To simplify training while retaining the structural alignment objective, we instead design Variance Concentration Loss (VCL), a distributional loss that encourages variance to concentrate along a small number of principal directions—regardless of data source. Let $R \in \mathbb{R}^{B \times d}$ denote the centered residual matrix. To ensure stable estimation of principal components, we collect residuals from a contiguous window of active transformer layers, based on prior observations that residual trajectories amplify and cluster within a small subset of layers. From the singular values $\{\sigma_j\}$ obtained via SVD $R = U\Sigma V^\top$, we define the auxiliary loss:

$$\mathcal{L}_{\text{VCL}} = -\frac{\sum_{j=1}^{k} \sigma_j^2}{\sum_{j=1}^{d} \sigma_j^2}$$

where $\gamma$ is a hyperparameter. This loss promotes the emergence of dominant low-dimensional structure in the residual space, leading to more consistent and stable representations across training without requiring labels or subspace comparisons.

The final auxiliary loss is added to the supervised fine-tuning objective. Formally, the total training loss becomes:

$$\mathcal{L}_{\text{total}} = \mathcal{L}_{\text{SFT}} + \gamma \cdot \mathcal{L}_{\text{VCL}}$$

where $\lambda$ controls the influence of the structural regularization.[4]

### 5.1   Selecting the Residual Window for PCA

To determine where our auxiliary loss will exert maximal influence, we first observe how residual norms evolve across layers. Specifically, we compute the $\ell_2$ norm of every residual vector and note an exponential growth trend with depth (Figure 9), consistent across models. This phenomenon arises from the additive update rule:

$$r_{i+1} = r_i + f(r_i)$$

---

[4]We provide the source code of at the anonymous link `https://anonymous.4open.science/r/CodeForPaper-3454`

where $r_i \in \mathbb{R}^d$ is the residual vector at layer $i$, and $f(r_i)$ is the learned update from attention and MLP modules. The squared norm evolves as:

$$\|r_{i+1}\|^2 = \|r_i\|^2 + 2\langle r_i, f(r_i)\rangle + \|f(r_i)\|^2$$

When the update $f(r_i)$ is approximately aligned with $r_i$, this leads to multiplicative growth:

$$\|r_{i+1}\| \approx \|r_i\| \cdot \sqrt{1 + \frac{\|f(r_i)\|^2}{\|r_i\|^2}}$$

which induces exponential scaling over depth: $\|r_i\| \sim a \cdot b^i$ for some $b > 1$. For example, in the LLaMA-3.2-1B model, the mean norm increases from 9.26 at layer 0 to 941.86 at layer 31. These results confirm that the residual stream follows an overarching amplification trend, indicating that interventions at earlier layers can effectively reshape its structure and providing a principled guide for choosing residual window $[l_1, l_2]$.

## 5.2 Experiments

**Experimental Setup And Evaluation Metrix**   We use the Llama-3.2-1B-SFT [13] model (trained via SFT on the `allenai/tulu-3-sft-mixture` dataset) as one of our baselines. We further compare against other false-refusal mitigation, including System Prompting, irected Representation Optimization (DRO) [40], Self-Contrastive Decoding (Self-CD) [28], and Vector Ablation strategies [34]. Evaluation is conducted on safety benchmarks and general capability tasks using Tülu 3 Evaluation Suite [17] . For safety evaluation, we include DAN, HarmBench, ToxiGen, WildGuard, JBB, and XSTest. For general capabilities, we report performance on MMLU, GSM8K, BBH, and CodexEval. In addition, we included OKTest, ORB-H and XSTest-H as False Refusal benchmarks following Wang [34]. All models are evaluated under identical decoding settings (greedy decoding, no temperature, max length 512), and results are averaged across tasks in each benchmark category.

**Main Results**   We evaluate the impact of our auxiliary loss on controlling instability induced by increasing proportions of safety data. As shown previously in Table 3, models trained without regularization suffer from growing distortion in residual geometry—measured via the alignment score along $\text{PC}_{ID}$—and rising false refusal rates as the ratio of safety examples increases. Evaluation results on larger models are provided in Appendix D.

| Model | Safety Benchmarks↑ | | | | False Refusal↑ | | | General Benchmarks↑ | | | |
|---|---|---|---|---|---|---|---|---|---|---|---|
| | DAN | Harmful | Toxigen | JBB | OKTest | ORB | XSTest | MMLU | GSM8K | BBH | CodexEval |
| Llama-3.2-1B-SFT | 0.78 | 0.74 | 0.90 | 0.76 | 0.53 | 0.76 | 0.51 | 0.42 | 0.50 | 0.25 | 0.24 |
| System Prompt | 0.79 | 0.75 | 0.95 | 0.77 | 0.71 | 0.65 | 0.58 | **0.45** | **0.52** | **0.27** | **0.34** |
| DRO | 0.80 | 0.72 | 0.92 | 0.81 | 0.63 | 0.71 | 0.68 | 0.39 | 0.49 | 0.24 | 0.23 |
| Self-CD | 0.76 | 0.81 | 0.91 | 0.83 | 0.77 | 0.426 | 0.78 | 0.38 | 0.50 | 0.26 | 0.23 |
| Vector Ablation | 0.84 | 0.80 | 0.97 | **0.91** | 0.67 | 0.447 | 0.58 | 0.37 | 0.51 | 0.25 | 0.24 |
| VCL(ours) | **0.89** | **0.841** | **1.000** | 0.86 | **0.76** | **0.87** | **0.86** | 0.42 | 0.51 | 0.26 | 0.25 |

Table 3: Benchmark results of Llama-3.2-1B

## 5.3 Hyperparameter Sensitivity: $l_1$, $l_2$, $k$ and $\gamma$

We conduct a sensitivity analysis to assess how the choice of the principal component cutoff $k$, the regularization weight $\gamma$, and the residual-window bounds $(l_1, l_2)$ affect model performance and residual geometry (see Section 5.1). Specifically, we vary $k$ in $\{1, 2, 4, 8\}$, $\gamma$ in $\{0.01, 0.1, 1.0, 2.0\} \times 50$, and $(l_1, l_2)$ corresponding to depths $[0.1, 0.3]$, $[0.3, 0.5]$, and $[0.5, 0.7]$. Each variant is evaluated on safety metrics such as the false refusal rate on XSTest and structural metrics such as variance concentration and cosine stability of leading PCs. Results show that the model is robust to $k$ in the 2–4 range but experiences degraded helpfulness when $\gamma$ is too large at small $k$, and that $\gamma = 1.0 \times 50$ yields the best overall performance. Among the residual-window settings, selecting $(l_1, l_2)$ to correspond to depth $[0.3, 0.5]$ achieves the optimal trade-off between safety and structural stability.

# 6 Conclusions, Limitations, and Future Work

**Limitations.** Our study focuses on the first $ID$ principal components, which capture the bulk of variance, but may overlook important structure present in the lower-variance directions. Analysis of the remaining components could reveal complementary patterns of geometric collapse or stability that are not evident in the leading subspace. Additionally, we apply a fixed $ID$ across all layers and prompts, which may not reflect layer- or context-specific intrinsic dimensions.

**Conclusions and Future Work.** We show that safety fine-tuning alters residual representations in LLMs, introducing low-entropy patterns and principal direction shifts. Our proposed loss improves refusal behavior without harming general capabilities. Future work will extend our analysis to broader settings, refine structural metrics beyond PCA, and develop more adaptive regularization schemes to balance safety and generalization.

# 7 Acknowledgements

This work was supported by the National Natural Science Foundation of China (Grant No. 62072052), the Foundation for Innovative Research Groups of the National Natural Science Foundation of China (Grant No. 61921003).

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

# A  Lexical Diversity Metrics

To quantify the lexical diversity and structural repetitiveness of safety versus general instruction data, we compute a set of surface-level metrics following the definitions in [16]. These include **token entropy ($H_1$, $H_2$, $H_3$)**, **mean segmental TTR (MSTTR)**, and the proportion of **unique $n$-grams**.

**Token Entropy.** We compute token entropy up to the third order to capture distributional characteristics:

$$ H_1 = -\sum_i p_i \log p_i, \quad H_2 = -\sum_i p_i (\log p_i)^2, \quad H_3 = -\sum_i p_i (\log p_i)^3, $$

where $p_i$ is the empirical probability of token $i$. Reporting $H_1$, $H_2$, and $H_3$ allows us to analyze both the mean entropy and its higher-order moments.

**Mean Segmental TTR (MSTTR).** To mitigate length sensitivity, MSTTR computes the TTR over fixed-length segments (here, 50 tokens), then averages across $N$ segments:

$$ \text{MSTTR} = \frac{1}{N} \sum_{j=1}^{N} \frac{|\text{Vocab}(j)|}{50}, $$

where $Vocab(j)$ is the set of unique tokens in segment $j$.

**Unique $n$-gram Ratio.** We compute the percentage of unique $n$-grams as:

$$ n\text{-gram Diversity} = \frac{\#\text{Unique } n\text{-grams}}{\#\text{Total } n\text{-grams}}. $$

In this work, we report results for $n=2$ (bigrams) and $n=3$ (trigrams), capturing local lexical variation in safety and general completions.

# B  Safety Data Selection Criteria

Constructing effective safety-aligned datasets for large language model training involves careful consideration of quality, diversity, and user privacy. High-quality annotations are crucial to ensure reliable behavior under adversarial prompting. Diversity is essential to cover a broad range of potential misuse cases and to prevent overfitting to narrow threat models. Privacy must also be strictly maintained, as safety prompts may involve sensitive or user-generated content. Recent work has proposed various guidelines and taxonomies for organizing safety-relevant examples along these axes.

**WILDJAILBREAK.** WILDJAILBREAK provides adversarial prompts collected via crowd-sourcing teams, targeting diverse harmful instruction styles. Each prompt is paired with a refusal completion generated under strict guidelines to ensure clarity and legal defensibility. This dataset contains over 100,000 safety-aligned completions; details of its collection pipeline are presented in Table 4.

**WILDGUARDMIX.** WILDGUARDMIX combines adversarial teaming and model-in-the-loop generation to produce challenging safety prompts. Completions are curated to cover a broad range of risk categories, from social engineering to illicit behavior, resulting in more than 100,000 refusal-type responses. See Table 4 for the full pipeline.

**TULU-3-SFT-MIXTURE.** The TULU-3-SFT-MIXTURE is a multi-domain instruction-tuning corpus with over 939,000 examples. We extract the safety subset—comprising refusal-type completions for sensitive or harmful queries—yielding more than 100,000 samples. Collection details appear in Table 4.

| Dataset | Completion Method | URL |
|---|---|---|
| WILDJAILBREAK [16] | Adversarial prompts collected via crowd-sourcing teams, paired with refusal completions generated under strict guidelines for clarity and legal defensibility ($> 100{,}000$ samples). | HF/WildJailbreak |
| WILDGUARDMIX [14] | Combines adversarial teaming and model-in-the-loop generation to produce challenging safety prompts; curated refusals across diverse risk categories ($> 100{,}000$ samples). | HF/WildGuardMix |
| TULU-3-SFT-MIXTURE [17] | Extracted safety subset of refusal-type completions from a multi-domain instruction corpus; over 100,000 safety-aligned examples. | HF/Tulu-3-SFT-Mixture |
| Control group | Randomly sampled 100,000 non-safety examples from the Tulu-3-SFT-Mixture-General subset as a control group. | HF/Tulu-3-SFT-Mixture |

Table 4: Completion collection methods and sample sizes for the four datasets used in our safety analysis. Collection pipelines are detailed in Table 4.

**Control group.** We randomly sample 100,000 non-safety examples from the TULU-3-SFT-MIXTURE-GENERAL subset as a control group for comparative analysis. The selection procedure is outlined in Table 4.

**On the limits of current data construction methods.** Despite the above efforts, we observe that many safety completions in public datasets follow highly uniform, templated patterns—e.g., "I'm sorry, but I can't...". This phenomenon is not merely a consequence of data construction pipelines, but a result of the task formulation itself. Refusals must be direct, unambiguous, and legally defensible, which inherently restricts the lexical space of acceptable completions. Consequently, even when the prompts are diverse, the completions tend to collapse into a few safe response modes.

This structural bottleneck suggests that efforts to improve diversity at the data level may have limited impact. Instead, we argue that the training objective should explicitly account for this asymmetry between prompt diversity and completion redundancy. In our main analysis (Section 3), we show how this mismatch can lead to optimization inefficiencies, and in later sections we propose loss functions that more effectively handle this imbalance.

## C  Top Trigram Frequencies in Safety and General Subsets

To further illustrate the lexical concentration in safety completions, we present the 25 most frequent trigrams in the safety and general subsets of the `Tülu 3 SFT Mixture`. All completions are tokenized using the `Llama-3.1-8B-Instruct` tokenizer. Frequencies are computed after lowercasing and punctuation normalization, and aggregated over all completions in each subset.

## D  Scaling to Larger Models

To examine whether our proposed structural loss continues to be effective at scale, we extend our evaluation to larger models. We apply the same fine-tuning configurations to a 7B-parameter variant and evaluate performance across both safety and general capability benchmarks. As shown in Table 6, the improvements observed in the 1B-scale experiments largely carry over. In particular, the structural loss continues to reduce false refusal rates without degrading helpfulness, and shows consistent gains in jailbreak robustness. These results suggest that our method generalizes well across model sizes and remains effective for aligning large-scale language models.

## E  Appendix: Stability Comparison between Cosine Similarity and FlowLens

This appendix compares the stability of two residual stream analysis tools: cosine similarity and FlowLens. While cosine similarity is widely used to measure angular relationships between token

| WILDJAILBREAK | WILDGUARDMIX | TULU-3-SFT-MIXTURE | CONTROL GROUP |
|---|---|---|---|
| ('. * *') 9191 | ('i can not') 4448 | ('i 'm sorry') 5062 | ('– – –') 1490 |
| ('. if you') 6737 | (', such as') 3146 | ('sorry , but') 5048 | (', such as') 679 |
| ('* * :') 5625 | ('. it is') 2660 | (''m sorry ,') 4980 | (', lo que') 578 |
| ('. i 'm') 5175 | ('. if you') 2534 | ('. i 'm') 4626 | ('. however ,') 518 |
| (', but i') 4980 | ('it is important') 2157 | ('i can not') 3021 | (': "" ') 516 |
| ('if you 're') 4901 | ('. however ,') 2128 | (', but i') 2975 | (', and the') 466 |
| ('. it 's') 4399 | ('is important to') 2128 | ('. if you') 2913 | ('" , "') 445 |
| ('i ca n't') 4298 | ('. i 'm') 2120 | (', i can') 2600 | ('sin embargo ,') 436 |
| ('it 's important') 4113 | (', it 's') 2057 | ('but i can') 1984 | ('. sin embargo') 418 |
| ('sorry , but') 4025 | ('. instead ,') 1993 | ('. however ,') 1950 | ('. además ,') 418 |
| (': * *') 3996 | ('. i can') 1909 | ('however , i') 1536 | ('. you can') 402 |
| (''s important to') 3957 | ('. it 's') 1807 | (', i do') 1360 | (', you can') 402 |
| ('i 'm sorry') 3643 | (', but i') 1800 | ('can not provide') 1161 | (', ya que') 374 |
| (''m sorry ,') 3624 | ('if you have') 1722 | ('i do not') 1110 | ('. * *') 352 |
| ('but i ca') 3492 | (', it is') 1658 | ('i can provide') 1079 | ('por ejemplo ,') 345 |

Table 5: Top-15 trigrams and their frequencies for each dataset: WILDJAILBREAK, WILD-GUARDMIX, TULU-3-SFT-MIXTURE, and Control group.

| Model | Safety Benchmarks↑ | | | | False Refusal↑ | | | General Benchmarks↑ | | | |
|---|---|---|---|---|---|---|---|---|---|---|---|
| | DAN | Harmful | Toxigen | JBB | OKTest | ORB | XSTest | MMLU | GSM8K | BBH | CodexEval |
| Llama-3.1-8B-SFT | 0.82 | 0.78 | 0.94 | 0.81 | 0.58 | 0.80 | 0.56 | 0.66 | 0.57 | 0.68 | 0.76 |
| System Prompt | 0.82 | 0.77 | 0.96 | 0.80 | 0.74 | 0.69 | 0.60 | **0.67** | **0.63** | **0.69** | **0.76** |
| DRO | 0.83 | 0.75 | 0.94 | 0.84 | 0.68 | 0.74 | 0.70 | 0.64 | 0.61 | 0.67 | 0.75 |
| Self-CD | 0.80 | 0.83 | 0.93 | 0.86 | 0.80 | 0.50 | 0.80 | 0.63 | 0.59 | 0.68 | 0.75 |
| Vector Ablation | 0.86 | 0.82 | 0.98 | **0.93** | 0.71 | 0.53 | 0.61 | 0.62 | 0.58 | 0.67 | 0.75 |
| VCL (ours) | **0.91** | **0.86** | **1.000** | 0.89 | **0.79** | **0.90** | **0.89** | 0.65 | 0.58 | 0.68 | 0.77 |

Table 6: Benchmark results of Llama-3.1-8B

vectors, it is sensitive to surface-level prompt perturbations. In contrast, FlowLens captures global geometric structure and demonstrates higher robustness.

**Experimental Setup.** We use 450 prompts from the XSTest benchmark [26], all of which are phrased as questions. Two prompt variants are constructed:

- **Punctuation group:** Prompts end with a question mark (e.g., "Can you describe this situation?").

- **No-punctuation group:** The same prompts with the question mark removed.

We use the LLaMA-3.1-8B-Instruct model. Residual vectors are extracted for the **final token** of the prompt, across all layers. Each residual vector has dimension $d = 4096$, and each prompt is analyzed independently (no batching). This final-token analysis approach is commonly used in prior works on residual stream interpretability [33, 3, 23].

**Cosine Similarity Analysis.** For each layer, we compute the average pairwise cosine distance between all final-token residuals. Let $X \in \mathbb{R}^{N \times d}$ be the residual matrix for $N$ prompts at a specific layer. The cosine distance is defined as:

$$\text{CosineDistance}(X) = \frac{1}{\binom{N}{2}} \sum_{1 \leq i < j \leq N} \left( 1 - \frac{X_i \cdot X_j}{\|X_i\| \|X_j\|} \right)$$

Results are shown in Figure 6. With punctuation, cosine similarity decreases from 1.0 to 0.6 across layers; without punctuation, it increases from 0.0 to 0.6. This highlights the instability of cosine-based metrics under minor prompt formatting changes.

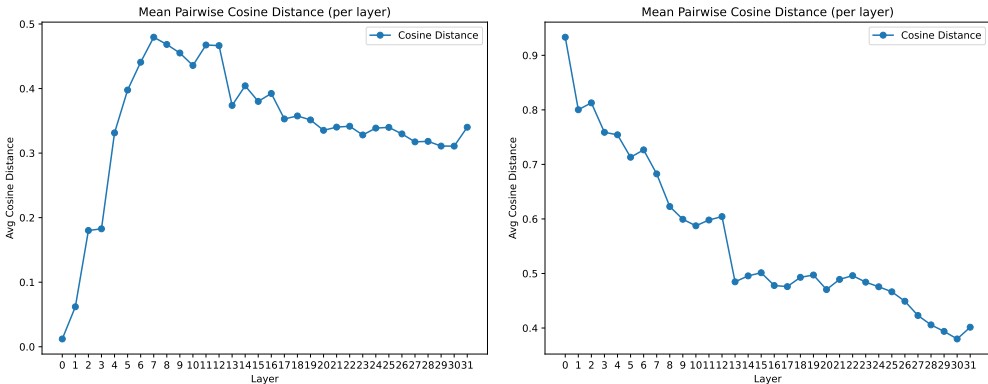

Figure 6: Cosine similarity across layers. Left: with punctuation; Right: without. Small changes cause dramatic shifts.

**Analysis with FlowLens.** We repeat the same experiment using FlowLens. For each prompt, we concatenate residuals from all layers into a single vector, and apply PCA to the resulting matrix of shape $(N, d \times L)$. Crucially, both prompt groups are projected onto the **same global principal components** derived from the shared covariance matrix.

As shown in Figure 7, projections onto PC1 exhibit consistent trends regardless of punctuation. This demonstrates that FlowLensis robust to superficial variations in prompt format, in contrast to cosine similarity, which relies on local angular differences.

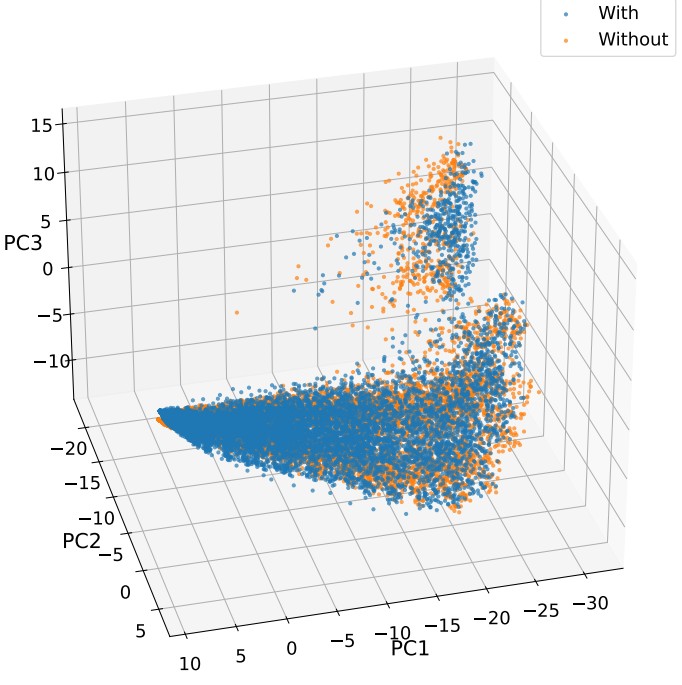

Figure 7: PCA projections. Trends remain stable despite surface-level changes.

**Discussion.** The analysis tools should be insensitive to the semantics but sensitive to the sentence structure of prompts. Current large language models exhibit robustness to input perturbations. However, robustness shown in output is not equal to the robustness in the residual stream. Thus we designed experiments to test stability of common analysis tools and FlowLens.

Prior work often analyzes cosine similarity between tokens or applies PCA layer by layer to study internal activations. However, both tools suffer from instability. Cosine similarity is highly sensitive to prompt formatting (Figure 6) and layerwise PCA often yields inconsistent principal axes across training stages or models due to basis rotation. These limitations motivate a more stable and comprehensive approach. Our findings suggest that FlowLensprovides a robust structural basis for analyzing the effects of safety fine-tuning.

# F    Additional PCA Projections Using FlowLens

We evaluate six instruction-tuned language models spanning multiple architectures and scales: LLaMA-3.2-1B-Instruct [13], LLaMA-3.1-8B-Instruct [13], LLaMA-2-7B-chat-hf [30], Qwen2.5-1.5B-Instruct [36], Phi-4-mini-instruct [1], and Gemma-3-4b-it [29]. As evaluation data, we use the TruthfulQA [20], a widely non-safety adopted dataset.

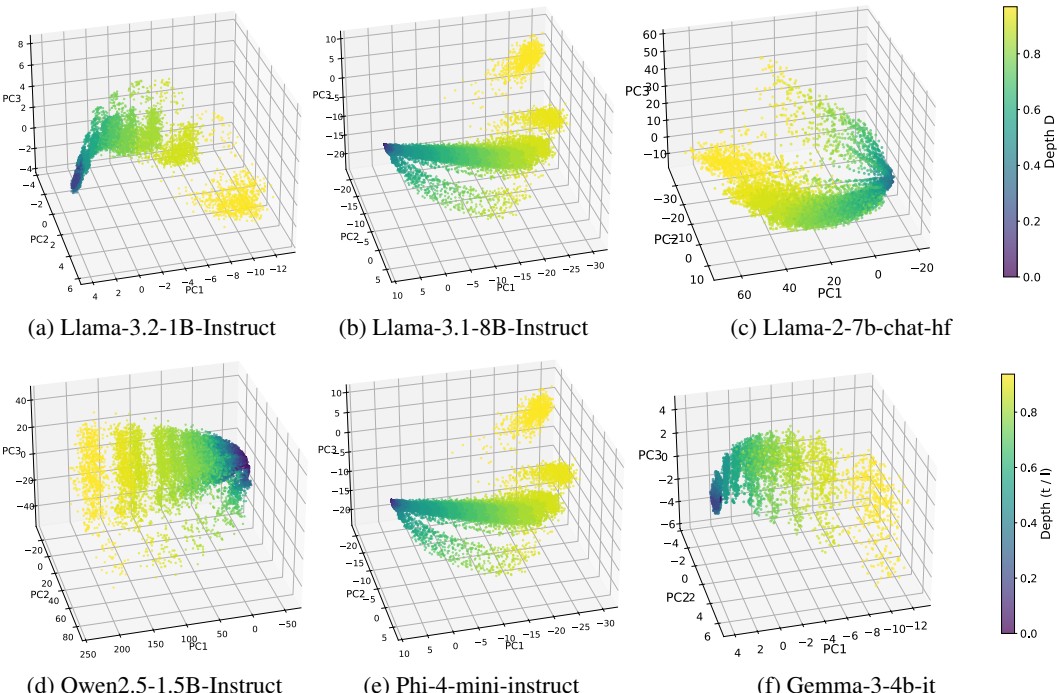

Figure 8: 3D PCA projections of residual trajectories using FlowLens for six instruction-tuned language models on the TruthfulQA dataset [20]. Each point represents the PCA-projected residual vector of the final token from one prompt, colored by its corresponding layer index (depth normalized to $[0, 1]$).

## F.1    Amplification and Dispersion Effects.

We examine semantic dispersion by measuring the mean distance of harmful prompt representations to their layerwise centroid (Figure 9). The results show exponential divergence, suggesting that safety fine-tuning spreads harmful representations further apart, possibly contributing to overgeneralized refusal patterns. We observe that the residual norm grows exponentially across layers, as expected from the additive nature of the residual connection. Figure 9 shows this trend across 50 prompts. Notably, this amplification effect magnifies the impact of instability at early layers, pushing distorted representations farther apart in deeper layers.

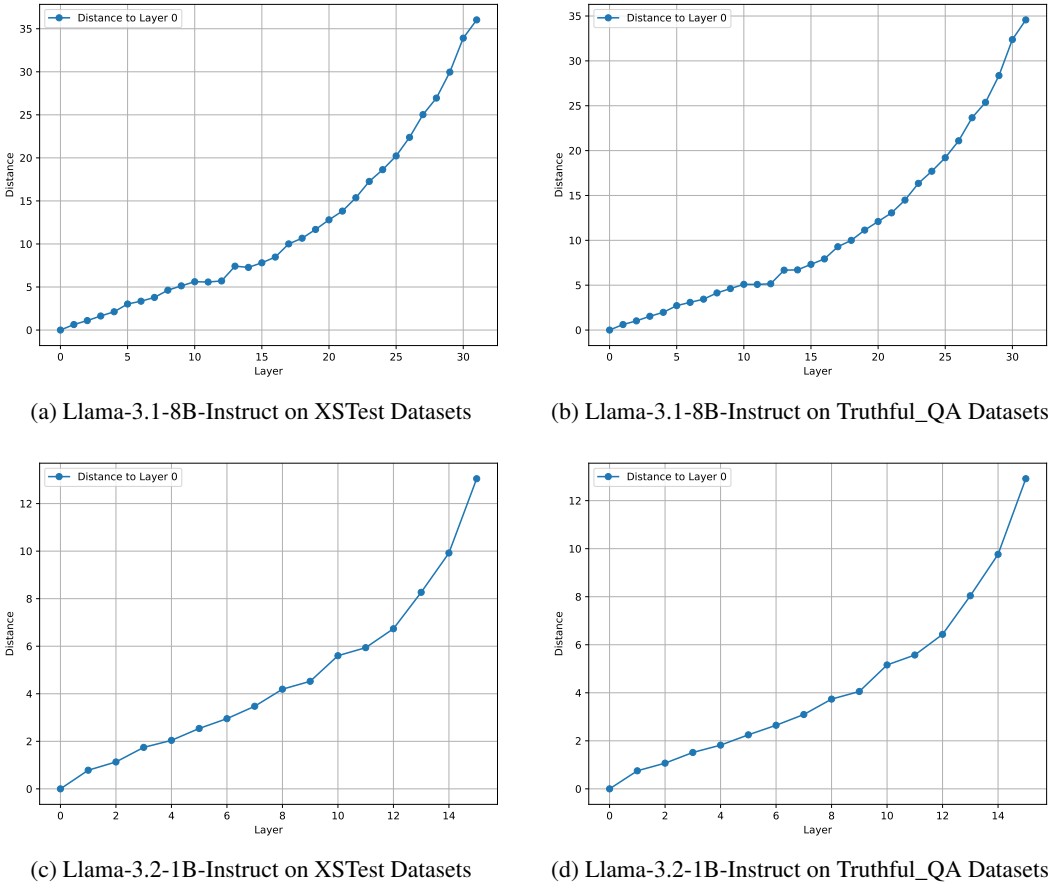

(a) Llama-3.1-8B-Instruct on XSTest Datasets

(b) Llama-3.1-8B-Instruct on Truthful_QA Datasets

(c) Llama-3.2-1B-Instruct on XSTest Datasets

(d) Llama-3.2-1B-Instruct on Truthful_QA Datasets

Figure 9: Mean distance to center for harmful prompts per layer across two model–dataset combinations.

# G   Additional Details

## G.1   Statistical Significance

To assess the variability of our results, we ran each experiment with three different random seeds (42, 100, 2025) and report mean ± standard deviation. For each benchmark metric $m$, we compute

$$\bar{m} = \frac{1}{3} \sum_{i=1}^{3} m_i, \quad \sigma_m = \sqrt{\frac{1}{3} \sum_{i=1}^{3} (m_i - \bar{m})^2}.$$

All tables and plots in the main text are now updated to display error bars corresponding to $\bar{m} \pm \sigma_m$.[5]

## G.2   Compute Resources

All experiments were conducted on a 8 NVIDIA A100-80G GPU.

- **Model fine-tuning**: Each run (LLaMA-3.2-1B-SFT) took approximately 4 hours wall-clock time, peak GPU memory usage 30 GB.

- **Residual analysis & PCA**: Approximately 2 hours per model, memory usage 8 GB.

- **Total compute**: ∼6 hours on one A100-80G; estimated carbon footprint: 0.3 kg $CO_2$.

---

[5]Details of seed selection and metric aggregation scripts are available in the anonymous code release.

## G.3 Broader Impacts

Our work carries several potential societal implications, with both positive and negative aspects. On the positive side, improved interpretability of safety-aligned large language models (LLMs) may accelerate trust in AI deployment, while our methods could guide more robust alignment procedures, thereby reducing over-cautious refusals. However, there are also risks of misuse: attackers might exploit insights into the residual stream to craft prompts that bypass safety filters, and the release of alignment diagnostics could enable adversarial fine-tuning to induce undesirable behaviors. To mitigate these risks, we recommend implementing gated access to the analysis tools, establishing clear usage guidelines, and actively monitoring for downstream misuse.

## G.4 Licenses for Existing Assets

- **WildJailbreak**, **WildGuardMix**, **Tulu-3-SFT-Mixture**: CC-BY 4.0 (as per `https://huggingface.co/datasets/xyz/LICENSE`).
- **LLaMA-3.1-8B** and **LLaMA-3.2-1B-SFT**: Meta Llama License v1.0 (`https://github.com/facebookresearch/llama/blob/main/LICENSE`).
- Our anonymous code release is under the MIT License.

