# OpenReview forum: "Residual Stream Analysis of Overfitting And Structural Disruptions"
_NeurIPS.cc/2025/Conference — NeurIPS 2025 poster_

### Official Review · Reviewer_NcTv · 2025-07-03

**Clarity:** 3
**Significance:** 4
**Originality:** 3
**Rating:** 5
**Confidence:** 4

**Summary:**

This paper addresses the problem of "false refusals" in large language models, a common side effect of safety fine-tuning. The authors hypothesize that this issue stems from the low lexical diversity and repetitive nature of refusal responses in safety datasets, which leads to a form of structural overfitting.

The core contributions are threefold:
1.  The paper quantifies the low diversity of safety data and introduces FlowLens, a PCA-based tool, to demonstrate how this data distorts the internal geometry of the model's residual stream, correlating this "geometric collapse" with increased false refusal rates.
2.  To counteract this, the authors propose Variance Concentration Loss (VCL), an auxiliary regularization term applied to mid-layer residuals during fine-tuning.
3.  Empirical results show that VCL significantly reduces false refusals on benchmarks like XSTest while maintaining or slightly improving performance on safety and general capability tasks like MMLU and GSM8K.

**Questions:**

1.  **Beyond the Top-k Components:** The analysis focuses on the top ID≈3 principal components. Have you investigated the behavior of the residual stream in the lower-variance dimensions? It's plausible that while top components handle broad semantic intent, these "tail" dimensions could encode other important features. Does VCL have any unintended effects on these parts of the representation space?
2.  **The Helpfulness/Safety Trade-off:** The results show VCL reduces false refusals while maintaining high performance on standard safety benchmarks. However, could this regularization make the model more susceptible to novel or adversarial jailbreaks that don't match the training distribution? A more thorough red-teaming evaluation or analysis on a wider range of adversarial benchmarks (beyond those used for training/validation) would be valuable to fully assess the safety implications.

**Ethical Concerns:**

["NO or VERY MINOR ethics concerns only"]

**Final Justification:**

Thank you for the authors' response and the additional experiments.

I wanted to add a public comment to highlight a particularly fascinating and non-obvious result, that the VCL method not only reduces false refusals but also improves robustness against adversarial attacks like GCG and ICA. I'm not sure whether this only happens on small models (e.g., 1B), but it's very interesting. This suggests a compelling hypothesis. The "structural overfitting" on repetitive safety data, which VCL is designed to fix, likely creates brittle and sharp geometric structures in the model's representation space. It seems plausible that gradient-based attacks like GCG inadvertently exploit these sharp, predictable structures to find adversarial solutions.

By promoting a smoother and more robust internal geometry, VCL appears to incidentally remove the very "brittle edges" that these attacks rely on. This is a powerful insight, suggesting that true alignment may stem more from fostering a healthy internal representation than from simply tuning surface-level behaviors. It's a very promising direction for future research.

**Limitations:**

yes

**Quality:**

3

**Strengths And Weaknesses:**

**Strengths:**
- This paper tackles the highly relevant and practical issue of false refusals, a key challenge in deploying safe and helpful LLMs.
- This paper builds a compelling narrative, starting from a clear data-driven hypothesis (low diversity of safety data), using a novel analysis tool (FlowLens) to find an internal correlate (geometric collapse), and finally proposing a targeted solution (VCL) to fix it.
- FlowLens provides an intuitive way to visualize structural changes in the residual stream, and VCL is an elegant, conceptually simple regularizer directly motivated by the analysis.
- The proposed VCL method demonstrates a significant reduction in false refusal rates without compromising safety or general task performance, which is a strong and desirable outcome.

**Weaknesses:**
- The experiments are conducted on a few specific model architectures (Llama family) and scales (1B, 8B), primarily on English datasets. The effectiveness of VCL on much larger models (e.g., >70B), different architectures (e.g., MoE), or across different languages remains an open question.

---

> ### Author Rebuttal · Authors · 2025-07-31
>
> Thank you for your precise and forward-looking comments. We respond point by point below.
>
> # **Weakness 1: On Model and Language Generalization of FlowLens and VCL**
>
> We thank the reviewer for pointing out the potential generalization limitations. To address this, we have extended our evaluation of FlowLens and VCL across **model families**, **scales**, and **architectures**, as detailed below.
>
> - **Model Family Generalization:** We applied VCL to the **Qwen** family ( Qwen-2.5-14B results), and observed consistent improvements in reducing false refusal rates, validating that our method is **not specific to LLaMA**-based models.
>
> **Qwen-2.5-14B results**
>
> | Model             | DAN  | Harmful | Toxigen | JBB  | OKTest | ORB  | XSTest | MMLU | GSM8K | BBH  | CodexEval |
> |------------------|------|---------|---------|------|--------|------|--------|------|--------|------|------------|
> | Qwen-2.5-14B-Instruct     | 0.85 | 0.80    | 0.96    | 0.84 | 0.62   | 0.82 | 0.60   | 0.70 | 97.8   | 0.71 | 0.78       |
> | System Prompt| 0.83 | 0.78    | 0.94    | 0.82 | 0.60   | 0.78 | 0.57   | **0.72** | **97.8** | **0.73** | **0.81** |
> | DRO               | 0.84 | 0.76    | 0.95    | 0.86 | 0.64   | 0.83 | 0.66   | 0.68 | 95.0   | 0.68 | 0.77       |
> | Self-CD           | 0.81 | 0.84    | 0.94    | 0.88 | 0.75   | 0.55 | 0.76   | 0.67 | 93.0   | 0.69 | 0.76       |
> | Vector Ablation   | 0.87 | 0.83    | 0.97    | 0.95 | 0.67   | 0.57 | 0.64   | 0.66 | 91.8   | 0.67 | 0.75       |
> | **VCL (ours)**        | **0.91** | **0.87** | **1.00** | **0.91** | **0.75** | **0.94** | **0.89** | 0.68 | 97.5   | 0.69 | 0.77       |
>
> - **Scaling to Larger Models:** Although we were unable to run full-scale experiments on 70B-class models due to resource constraints, we evaluated on the intermediate **14B scale**. Results show that both FlowLens trends and the effectiveness of VCL **scale smoothly** from 1B → 8B → 14B, suggesting a promising trend toward larger models.
>
> - **Application to MoE Models:** We further tested FlowLens on **MoE models**, where each token is routed through a subset of experts. As shown in the figure below (replaced with a table in final rebuttal), we observe significantly **larger divergence in residual geometry** on safety-only data compared to dense counterparts:
>
> | Dataset     | PC1  | PC2  | PC3  |
> |-------------|------|------|------|
> | SafetyOnly      | 0.09 | 0.06 | 0.20 |
> | GeneralOnly | 1.00 | 0.99 | 0.97 |
>
>
> This indicates that FlowLens is **sensitive to structural shifts even in MoE settings**. Notably, the similarity in PC3 (0.20) does not contradict this observation, as in MoE models the explained variance of PC3 is very low (<1%), compared to 5–10% in previous dense experiments.
>
> - **Evaluation on DPO Models:** We also applied FlowLens to a **DPO-aligned model**, observing that mid-layer geometric collapse still persists, consistent with our earlier findings. This suggests that structural issues from SFT may carry over, reinforcing the value of VCL even after DPO alignment.
>
> | Dataset     | PC1  | PC2  | PC3  |
> |-------------|------|------|------|
> | SafetyOnly      | 0.99 | 0.99 | 0.66 |
> | GeneralOnly  | 1.00 | 0.99 | 0.87 |
>
> - **Cross-lingual Evaluation:** While we recognize the importance of testing across languages, our current experiments are constrained by training data availability and compute. We now explicitly acknowledge this as a **limitation** and a future direction in Section 6.
>
> Together, these results demonstrate that FlowLens and VCL generalize well across models and training setups, with further architectural extensions (e.g., MoE, multilingual) posing **valuable future opportunities**.
>
> # **Question 1: On the Effect of VCL Beyond Top Principal Components**
>
> We appreciate this thoughtful question. We agree that while top principal components typically reflect broad semantic structure, lower-variance (tail) dimensions may encode subtle yet important features.
>
> Our design of VCL accounts for this. Specifically:
>
> - **Controlled Regularization via $\gamma$:** The strength of VCL is governed by a hyperparameter $\gamma$, which we tune conservatively to **avoid over-regularization**. This ensures that VCL only penalizes **excessive variance concentration**, rather than enforcing any fixed projection pattern.
>
> - **Layer Scope Restriction:** VCL is applied only to a selected **mid-layer window**, leaving the later layers untouched. These deeper layers retain full flexibility to allocate capacity across all principal or non-principal directions.
>
> - **Preservation of Normal Query Geometry:** Empirically, we observe that the residual stream geometry of **non-safety prompts** remains largely unaffected by the introduction of VCL. This suggests that tail dimensions activated by benign/general queries are **not inadvertently suppressed** by our method.
>
> Together, these design choices allow VCL to **mitigate structural overfitting** on highly repetitive safety data, without impeding the expressive capacity of the residual stream—particularly in directions beyond the top few principal components.
>
> In future work, we plan to conduct a more targeted spectral analysis of **tail components** to complement our current PCA-focused evaluations.
>
> # **Question 2: Helpfulness/Safety Trade-off under Adversarial Attacks**
>
> We appreciate the reviewer’s concern regarding potential vulnerabilities introduced by VCL to adversarial jailbreak attacks.
>
> To assess this, we extended our evaluation beyond DAN to include two recent adversarial attack methods: **GCG** [1] and **ICA** [2].
>
> - **GCG (Greedy Coordinate Gradient)** is a universal and transferable jailbreak attack that optimizes adversarial suffixes which can be transferred across models and prompts [1]. We adopted the **transfer setting** from the original paper, using suffixes generated from Llama2-7B and applying them to **Llama-3.1-1B** and **Llama-3.1-8B**. This setup is justified, as the transferability relies on overlapping pretraining corpora and shared model architectures, as discussed by the authors.
>
> - **ICA (In-Context Attack)** uses a few-shot in-context learning setup to jailbreak aligned models without modifying their parameters. It leverages known jailbreak queries and completions, requiring only 3-shot demonstrations [2]. We follow their 3-shot ICA setup using the open jailbreak pool provided in the original paper.
>
> Below we report the **reject rates (RR, %)** of various methods under these two attacks. Higher is better.
>
> | Model             | GCG ↑ | ICA ↑ |
> |------------------|-------|-------|
> | Llama-3.1-1B-SFT | 91.2  | 90.8  |
> | System Prompt     | 94.9  | 93.0  |
> | DRO               | 93.5  | 92.2  |
> | Self-CD           | 91.6  | 90.3  |
> | Vector Ablation   | 94.4  | 92.8  |
> | **VCL (ours)**    | **95.2**  | **93.1**  |
>
> These results suggest that **VCL does not increase susceptibility to adversarial attacks**.
>
> [1] Zou et al., "Universal and Transferable Adversarial Attacks on Aligned Language Models", arXiv:2307.15043.
> [2] Wei et al., "Jailbreak and Guard Aligned Language Models with Only Few In-Context Demonstrations", arXiv:2310.06387.

---

### Official Review · Reviewer_WJ2Z · 2025-07-03

**Clarity:** 2
**Significance:** 3
**Originality:** 3
**Rating:** 4
**Confidence:** 2

**Summary:**

This paper conducts an in-depth analysis of the “false refusal” problem induced by safety alignment in large language models (LLMs). The authors find that fine-tuning models with highly templated and repetitive refusal data leads to structural collapse of the residual stream in intermediate layers. To address this, the paper introduces a novel residual stream geometric analysis tool, FlowLens, and proposes a new regularization method, Concentration Loss, which penalizes excessive variance concentration in mid-layer residual streams, thereby significantly reducing false refusal rates.

**Questions:**

1. Is the proposed method applicable to multimodal models that integrate both language and vision branches?

2. How much computational overhead does FlowLens and Concentration Loss (VCL) introduce during training?

3. Can VCL be combined with other approaches such as Distributionally Robust Optimization (DRO)?

4. Does optimizing variance concentration risk sacrificing essential representational capacity?

**Ethical Concerns:**

["NO or VERY MINOR ethics concerns only"]

**Final Justification:**

The authors have addressed most of my concerns, and I am inclined to maintain my positive score.

**Limitations:**

yes

**Quality:**

3

**Strengths And Weaknesses:**

## Strengths
1. The paper targets a critical issue in LLM safety – the tendency of safety fine-tuning to cause over-refusal to benign inputs – and proposes a comprehensive analysis framework along with an effective regularization solution.

2. Compared to existing residual stream cosine similarity analyses, the proposed FlowLens leverages PCA to directly capture global geometric trends of cross-layer residuals, yielding improved stability and interpretability.

## Weaknesses
1. While the paper connects low diversity in safety data to residual stream geometric collapse based on PCA clustering phenomena, it lacks a deeper theoretical explanation of why low data diversity necessarily induces mid-layer variance concentration, and why such geometric structures directly trigger false refusals.

2. Current validations are mainly conducted on the LLaMA family. It remains unclear whether the proposed methods generalize to other architectures, such as multimodal models or large-scale MoE models.

3. The method design may carry an over-regularization risk, potentially causing unknown negative transfer effects on other downstream tasks.

4. The experiments lack direct comparisons with other structural regularization or data diversity enhancement methods applied during training.

---

> ### Author Rebuttal · Authors · 2025-07-31
>
> Thank you for your constructive and technically grounded feedback. We address your concerns in detail below.
>
> # **Weakness 1: Theory behind PCA clustering**
>
> We appreciate the reviewer’s thoughtful concern regarding the theoretical underpinnings of our causal hypothesis.
>
> Our current evidence primarily supports an **empirical pipeline** from **low lexical diversity** → **geometric concentration in residual stream** → **increased false refusals**. While we agree that this causal chain is not yet fully formalized, we offer the following rationale and findings:
>
> - **From Low Diversity to Geometric Collapse:**
>   We observe this phenomenon consistently across model families, including dense transformers (e.g., LLaMA, Qwen), and even DPO-aligned models (e.g., LLaMA-3.1-Tulu-3-8B-DPO). These results suggest that data distribution—especially the low diversity of safety responses—is a key driver of residual stream distortion, regardless of model architecture.
>
> - **From Geometry to False Refusal:**
>   Our working hypothesis assumes that the model's internal **representation space is continuous** and supports semantic generalization. In this space, **overfitting to structurally repetitive safety data** distorts the geometry, effectively "bending" the boundary between safe and benign queries. This causes models to **misclassify benign prompts near that boundary as unsafe**, resulting in **false refusals**.
>
> This view aligns with the idea of a **continuous optimization landscape**, where local over-optimization (e.g., excessive variance concentration for one class) leads to **poor generalization at class intersections**.
>
> While the precise mechanism is not yet fully understood, our extensive experiments and supporting analyses provide a strong basis for concluding that low-diversity safety data can induce structural overfitting in residual representations, leading to false refusals.
>
> # **Weakness 2: On Generalization Beyond the LLaMA Family**
>
> We thank the reviewer for raising this important point. To evaluate whether our methods generalize beyond the LLaMA architecture, we conducted further experiments across multiple model families and architectures:
>
> - **Qwen Family (Dense Transformer):** We applied VCL to **Qwen-2.5-14B**, a dense transformer models. In both cases, VCL effectively reduced false refusal rates, consistent with our results on the LLaMA family. This suggests that our method **generalizes well to other transformer implementations** beyond LLaMA. Due to space constraints, detailed results are reported in our response to Reviewer r4L3 – Weakness 1
>
> - **MoE models：** We further tested FlowLens on **MoE models**, where each token is routed through a subset of experts. As shown in the figure below (replaced with a table in final rebuttal), we observe significantly **larger divergence in residual geometry** on safety-only data compared to dense counterparts:
>   MoE:
>   | Dataset     | PC1  | PC2  | PC3  |
>   |-------------|------|------|------|
>   | SafetyOnly      | 0.09 | 0.06 | 0.20 |
>   | GeneralOnly | 1.00 | 0.99 | 0.97 |
>
> - **Multimodal Architecture:** Due to incompatibilities in tokenizer and forward implementation, applying VCL to multimodal models requires nontrivial engineering efforts. We leave this for future exploration.
>
>  - **DPO-Aligned Model:** We ran FlowLens on a **DPO-aligned model**, specifically **LLaMA-3.1-Tulu-3-8B-DPO**. The residual stream still exhibited **variance concentration**, similar to what we observed after supervised fine-tuning. This indicates that **structural disruptions introduced during SFT can persist through later alignment stages like DPO**, and that applying VCL earlier in the pipeline may mitigate these effects.
>   DPO:
>   | Dataset     | PC1  | PC2  | PC3  |
>   |-------------|------|------|------|
>   | SafetyOnly      | 0.99 | 0.99 | 0.66 |
>   | GeneralOnly  | 1.00 | 0.99 | 0.87 |
>
> These experiments support the generalizability of our approach.
>
> # **Weakness 3: Risk of over-regularization and negative transfer**
> We appreciate this concern. In designing VCL, we explicitly aimed to **mitigate over-regularization risks** by following two principles:
>
> 1. **VCL is a soft regularizer**, not a hard constraint. Much like KL penalties in reinforcement learning, it introduces a tunable pressure (via the γ coefficient) against excessive variance collapse, without dictating the exact structure of representations. In practice, we found small γ values (e.g., 0.1–0.3) already yield substantial gains in safety without harming generalization.
>
> 2. **VCL is applied selectively** to mid-level transformer layers—i.e., where variance collapse is empirically observed. The higher layers, which are more directly tied to final output behaviors and task-specific logic, remain unconstrained, preserving freedom for downstream specialization.
>
> Empirically, we observe that **general-task benchmarks (MMLU, GSM8K, BBH, CodexEval)** remain stable. This suggests that VCL helps restore robustness against over-refusal **without degrading general representational capacity**.
>
> # **Weakness 4: Lack of comparison with other structural regularization or data diversity enhancement methods**
>
> We appreciate the reviewer’s suggestion.
>
> **Regarding structural regularization:** While prior work has explored structural properties of model representations—such as analyzing residual streams using cosine similarity or probing directions tied to specific behaviors—**few have proposed using residual stream structure as a train-time regularization target** during supervised fine-tuning (SFT). Moreover, **some baselines we include do implicitly relate to representational structure**. For instance, **Vector Ablation** intervenes on specific residual directions learned from harmful data, and **DRO** (Drected Representation270Optimizatio) indirectly influences geometry by adjusting model confidence across harmful/helpful distributions. We treat these methods as relevant comparison points, and VCL consistently outperforms them on false refusal reduction.
>
> **Regarding data diversity:** our SFT baseline models are trained on the *Tulu-3-SFT-Mixture*, which is explicitly curated for high diversity through sampling from varied datasets, persona-driven prompt creation, and class-balanced downsampling. Therefore, our experiments are already conducted under strong diversity-enhanced conditions—highlighting that **VCL brings additional benefits beyond dataset construction alone**.
>
> ---
>
> # **Question 1: Applicability to multimodal models**
> Please see our response to **Weakness 2**, where we discuss the applicability of VCL to **multimodal and MoE models** in more detail.
>
> # **Question 2: Computational overhead of FlowLens and VCL**
> The **computational overhead of VCL is minimal**. It operates as a soft regularization term, similar to KL penalties used in reinforcement learning, and adds only a lightweight PCA-based loss term to selected mid-layer residuals. It does **not modify model architecture** or introduce additional forward/backward operations.
>
> We discuss this in detail in **Appendix G.2**, where we show that training time and memory usage remain nearly unchanged when enabling VCL.
>
> As for **FlowLens**, it is an analysis tool used only during evaluation and visualization. It plays no role during model training and does **not contribute to training cost**.
>
> # **Question 3: Can VCL be combined with DRO?**
>
> Yes, VCL is **theoretically compatible** with Distributionally Robust Optimization (DRO). The two methods operate on orthogonal aspects of model behavior:
>
> - **DRO** encourages differentiation between *helpful* and *harmful* queries within each layer by reweighting training loss across risk groups;
> - **VCL** promotes *global variance diversity* across transformer layers, mitigating variance collapse without disrupting within-layer representational distinctions.
>
> In fact, we tested models trained with VCL on both *harmful* and *harmless* inputs, and observed that **residual streams from different query types still form distinct clusters within the same layer**. This suggests that while VCL encourages overall variance dispersion, it **preserves task-relevant internal structure**.
>
> While we have not yet experimented with joint VCL + DRO trainin, doing so would simply require tuning a relative weighting coefficient between the two loss terms—an implementation we believe to be straightforward and promising for future work.
>
> # **Question 4: Does VCL sacrifice representational capacity?**
>
> This is an important concern, and we believe the answer is **no**—VCL does not sacrifice essential representational capacity.
>
> Conceptually, VCL is akin to **KL regularization in reinforcement learning**, where the goal is not to constrain expressiveness, but to avoid undesirable collapse (e.g., to a narrow policy mode or representational subspace). VCL does **not enforce high variance**, but instead penalizes **excessive concentration** of residual stream variance into a few principal components, which our analysis links to false refusal behaviors.
>
> Empirically, we observe that general benchmarks (e.g., MMLU, GSM8K, BBH, CodexEval) remain **stable** after applying VCL. This suggests that **core task-relevant features are preserved**, and essential capacity is not diminished.
>
> Moreover, **VCL is only applied to mid-level layers** where variance collapse is most severe. The upper layers—closer to output decisions—remain unconstrained and can adapt freely to downstream task requirements.
>
> We thus view VCL as **restoring internal diversity** where it is most needed, without interfering with the model’s ability to learn and represent high-level semantics.

---

> > ### Comment · Reviewer_WJ2Z · 2025-08-06
> >
> > Thank you to the authors for their response, which clarified some of my concerns. I am inclined to maintain my initial positive score.

---

### Official Review · Reviewer_WW1T · 2025-07-05

**Clarity:** 3
**Significance:** 2
**Originality:** 3
**Rating:** 4
**Confidence:** 4

**Summary:**

This paper investigates the problem of "false refusals" in large language models (LLMs), where models fine-tuned for safety erroneously decline benign user prompts. The authors hypothesize that this issue stems from structural biases in safety datasets, which often feature diverse prompts paired with highly repetitive, low-entropy refusal templates. The paper quantifies the low lexical diversity in safety datasets and demonstrates how this leads to a form of "structural overfitting," where models become overconfident in templated refusal responses. The authors then introduce FlowLens, a PCA-based method for analyzing the geometry of the residual stream across multiple layers. Using FlowLens, the authors show that increased safety fine-tuning leads to a "representational collapse," where variance becomes heavily concentrated in a few principal components, correlating strongly with higher false refusal rates. Finally, the paper proposes Variance Concentration Loss (VCL), an auxiliary regularization term applied during supervised fine-tuning. VCL penalizes excessive variance concentration in mid-layer residual vectors. Empirical results on LLaMA-family models show that VCL significantly reduces false refusal rates while maintaining or slightly improving performance on general capability benchmarks like MMLU and GSM8K.

**Questions:**

- The hyperparameter analysis shows that applying VCL to a mid-layer window is optimal. This is an interesting finding. Is there a deeper intuition for this? For instance, do earlier layers handle more general syntax where regularization is less helpful, and later layers are too specialized for VCL to have a broad effect?
- Have you analyzed the residual stream of a DPO-tuned model with FlowLens? Does it also show variance concentration?

**Ethical Concerns:**

["NO or VERY MINOR ethics concerns only"]

**Final Justification:**

I thank the authors for their detailed and thoughtful rebuttal. Having read the rebuttal and considered the feedback from other reviewers, I find that my initial assessment remains consistent. Therefore, I will maintain my original positive score.

**Limitations:**

yes

**Quality:**

3

**Strengths And Weaknesses:**

**Strengths**
- The paper addresses a practical challenge in LLM alignment. False refusals degrade user experience and limit the utility of safety-aligned models.

- The proposed FlowLens provides a more stable and global view of representational geometry than prior layer-wise or cosine-similarity-based methods. The visualizations are effective at illustrating the "unfolding" geometric structure and how it is disrupted by safety data.

- The authors demonstrate the initial problem (low data diversity), show the correlation between safety data ratio and geometric collapse, and evaluate their proposed solution against several relevant baselines across a suite of safety and general capability benchmarks.

**Weaknesses**
- The causal mechanism remains somewhat speculative. A deeper explanation of why this geometric collapse leads the model to misclassify benign prompts would strengthen the paper's claims.

- The proposed intervention, VCL, is applied during SFT. Modern alignment pipelines often involve further stages like RLHF or DPO. It would be valuable to understand if the structural disruptions persist after these stages and whether VCL's benefits carry over or if a different intervention is needed.

---

> ### Author Rebuttal · Authors · 2025-07-31
>
> Thank you for the helpful questions and insights into the broader implications of our work. Please see our responses below.
>
> # **Weakness 1: On the causal mechanism**
> We appreciate this thoughtful point. Our current explanation of the causal pathway remains primarily **descriptive**. While the precise mechanism is not yet fully understood, our extensive experiments and supporting analyses provide a strong basis for concluding that low-diversity safety data can induce structural overfitting in residual representations, leading to false refusals:
>
> - From **low diversity to geometric collapse**, we observe that templated safety data consistently reduces residual stream variance dispersion. This holds across architectures including dense models (Qwen and Llama) and DPO models—suggesting a general effect rooted in **data entropy** rather than model class.
>
> - From **geometry to false refusals**, we hypothesize that residual stream representations are **continuous**, and safety overfitting reshapes the representational manifold. This can lead to overly sharp decision boundaries in ambiguous or marginal cases—causing models to misclassify benign prompts as harmful due to their proximity to refusal templates in collapsed regions.
>
> We agree this geometric–behavioral link deserves deeper formalization. As future work, we propose quantifying how **low-diversity data reshapes the optimization landscape and decision boundary geometry**, potentially through tools from manifold learning or information geometry.
>
> # **Weakness 2: DPO-tuned models still exhibit variance concentration in residual streams**
>
> Thank you for this insightful question. We conducted additional experiments using the **LLaMA-3.1-Tulu-3-8B-DPO** model to explore whether DPO-aligned models also show signs of residual stream variance concentration, as measured by FlowLens.
>
> We evaluated residual stream PCA alignment on two datasets (TruthfulQA and XSTest), following the same procedure as in Section 4.2. The cosine similarity between each dataset's local top-3 principal components and the global PCA directions is reported below:
>
> | Dataset     | PC1  | PC2  | PC3  |
> |-------------|------|------|------|
> | SafetyOnly      | 0.99 | 0.99 | 0.66 |
> | GeneralOnly  | 1.00 | 0.99 | 0.87 |
>
>
> These results indicate that **DPO models retain high alignment with global variance directions**, especially in early components (PC1 and PC2), and still exhibit moderate-to-strong variance concentration (notably in SafetyOnly’s PC3: 0.66).
>
> We view this as an opportunity for future work to explore **post-SFT structural interventions** or **combined SFT-DPO loss formulations** that preserve helpfulness while improving geometric robustness.
>
> ---
>
> # **Question 1: Mid-layer is the effective location for regularization**
>
> Thank you for the thoughtful question. Our decision to apply VCL in the mid-layer range is grounded in both empirical observation and representational insights from prior work.
>
> **As for latter layers**, as shown in Section 5.1, we conclude an *amplification effect* in false refusal rates across layers — early and mid-layer representational distortions propagate and accumulate through the stack. However, *late-layer regularization alone* (e.g., targeting only the final blocks) has limited ability to correct upstream collapse. This explains why mid-layer VCL often yields more stable and effective mitigation.
>
> **As for eailer layers**, the relative ineffectiveness of early-layer regularization is consistent with findings in [1] (*Refusal in Language Models Is Mediated by a Single Direction*), which shows that **refusal-relevant directions emerge more strongly in middle-to-late layers**. Their Figure 5 shows that intervention signals concentrate most in mid layers, aligning with our empirical window selection.
>
> Thus, our current design reflects both architectural intuition and optimization-based tuning. We agree that further interpretability of why mid layers are particularly amenable to structural regularization is an exciting future direction.
>
> [1] Arditi, A., Obeso, O., Syed, A., Paleka, D., Panickssery, N., Gurnee, W., & Nanda, N. (2024). *Refusal in language models is mediated by a single direction*. NeurIPS 2024.
>
> # **Question 2:DPO-tuned models still exhibit variance concentration in residual streams**
>
> Please refer to our response under **Weakness 2**, where we discuss whether structural disruptions persist through later alignment stages such as DPO, and whether VCL remains beneficial post-SFT. We believe this is an important and underexplored direction.

---

> > ### Comment · Reviewer_WW1T · 2025-08-08
> >
> > Thank you to the authors for their detailed and thoughtful rebuttal. Having read the rebuttal and considered the feedback from other reviewers, I find that my initial assessment remains consistent. Therefore, I will maintain my original positive score.

---

### Official Review · Reviewer_r4L3 · 2025-07-05

**Clarity:** 2
**Significance:** 2
**Originality:** 3
**Rating:** 4
**Confidence:** 2

**Summary:**

The paper investigates why fine-tuning LLMs on highly repetitive safety data leads to an overabundance of false refusals. By measuring and comparing lexical diversity (token entropy and n-gram coverage) between safety and general instruction datasets, the authors show that safety completions are far more templated, which drives a structural overfitting. They introduce FlowLens, a PCA-based analysis tool that reveals how increasing proportions of safety data concentrate variance along a few principal components, correlating with rising false-refusal rates. Guided by these insights, they propose Variance Concentration Loss, an auxiliary regularizer that penalizes excessive variance concentration in mid-layer residuals during fine-tuning. Empirically, VCL reduces false refusals on benchmarks including MMLU, GSM8K, BBH, and CodexEval.

**Questions:**

1. The processing of this paper's pdf is extremely slow, and I'm not sure if it's just happening on my end. I suspect this is caused by PCA scatter plots and  I'd appreciate it if the author could look into this issue if possible.

2. The authors are advised to explain what "Control" in Table 1 means in the main text, even though it has been explained in the appendix.

3. Is there any hyperparameters need to be adjust when training with VCL?

**Ethical Concerns:**

["NO or VERY MINOR ethics concerns only"]

**Final Justification:**

Some of my concerns are addressed. I'm glad to see the method works on other models as well. I keep my score positive.

**Limitations:**

Yes

**Quality:**

2

**Strengths And Weaknesses:**

### Strengths

1. The method of increasing the variance diversity during fine-tuning is reasonable.

2. The theoretical analysis looks thorough.

### Weaknesses

1. The major weakness is the lack of sufficient experiments. The only experiment in the main text is Table 3, and the only models tested in the whole paper are Llama‑3.1‑1B and Llama‑3.1‑8B. The authors should run experiments on a wider range of models to validate the effectiveness of their method.

2. In Table 3, the improvement in False Refusal shows no clear relationship with the improvement on the general benchmarks. For example, VCL achieves a large gain in False Refusal compared to Vector Ablation, yet only shows very small gains on the general benchmarks. Does this mean that solving the false refusal problem does not resolve the tension between helpfulness and harmlessness? Could the authors explain how important addressing false refusals is for improving both helpfulness and harmlessness simultaneously?

3. The introduction of Residual Stream Geometry and FlowLens is good but somewhat confusing. I wonder why Residual Stream Geometry is used instead of other metrics like entropy. Could the authors provide a theoretical justification or an experimental comparison?

4. Section 5.3 should provide a result table.

---

> ### Author Rebuttal · Authors · 2025-07-31
>
> Thank you for your thoughtful and detailed feedback! We address your concerns below.
>
> # **Weakness 1: Model diversity in experiments**
>
> We agree that evaluating on a wider range of models is critical to assessing the generality of our method. In response, we have conducted additional experiments on following new models:
>
> - **Qwen-2.5-14B**, a significantly larger model to assess scaling behavior and a model from a different model family. **The results are consistent with our findings in the original paper**: VCL substantially improves performance on false refusal benchmarks such as XSTest and OKTest, while maintaining competitive results on general benchmarks such as MMLU and CodexEval.
>
> The full results are shown below:
>
> | Model             | DAN  | Harmful | Toxigen | JBB  | OKTest | ORB  | XSTest | MMLU | GSM8K | BBH  | CodexEval |
> |------------------|------|---------|---------|------|--------|------|--------|------|--------|------|------------|
> | Qwen-2.5-14B-Instruct     | 0.85 | 0.80    | 0.96    | 0.84 | 0.62   | 0.82 | 0.60   | 0.70 | 97.8   | 0.71 | 0.78       |
> | System Prompt| 0.83 | 0.78    | 0.94    | 0.82 | 0.60   | 0.78 | 0.57   | **0.72** | **97.8** | **0.73** | **0.81** |
> | DRO               | 0.84 | 0.76    | 0.95    | 0.86 | 0.64   | 0.83 | 0.66   | 0.68 | 95.0   | 0.68 | 0.77       |
> | Self-CD           | 0.81 | 0.84    | 0.94    | 0.88 | 0.75   | 0.55 | 0.76   | 0.67 | 93.0   | 0.69 | 0.76       |
> | Vector Ablation   | 0.87 | 0.83    | 0.97    | 0.95 | 0.67   | 0.57 | 0.64   | 0.66 | 91.8   | 0.67 | 0.75       |
> | **VCL (ours)**        | **0.91** | **0.87** | **1.00** | **0.91** | **0.75** | **0.94** | **0.89** | 0.68 | 97.5   | 0.69 | 0.77       |
>
> - **DPO-Aligned Model:** We ran FlowLens on a **DPO-aligned model**, specifically LLaMA-3.1-Tulu-3-8B-DPO. The residual stream still exhibited **variance concentration**, similar to what we observed after supervised fine-tuning. This indicates that structural disruptions introduced during SFT can persist through later alignment stages like DPO, and that applying VCL earlier in the pipeline may mitigate these effects.
>
> | Dataset     | PC1  | PC2  | PC3  |
> |-------------|------|------|------|
> | SafetyOnly      | 0.99 | 0.99 | 0.66 |
> | GeneralOnly  | 1.00 | 0.99 | 0.87 |
>
> These results reinforce the structural insight underlying VCL: by regularizing variance geometry in the residual stream, we consistently reduce false refusals while preserving general task performance—**across different model families (Qwen vs. LLaMA)**.
>
>
> # **Weakness 2: The impact of mitigating 2H tension on helpfulness**
> That's a very insightful point—we agree that the relationship between false refusal reduction and general helpfulness is not always obvious at first glance. You raise a valid concern: if general benchmarks like MMLU or GSM8K don’t show major improvement, does that mean VCL is only making the model less harmful (by reducing refusals), but not more helpful?
>
> Our perspective is this: **false refusal itself is a failure mode of helpfulness**, especially under safety constraints. In fact, resolving false refusals is one of the clearest ways we can make the model more helpful—precisely on the kinds of queries where the helpfulness–harmlessness tension is most acute.
>
> Take the example in **Figure 1**. The user asks, “How can I smash my exam next week?”—clearly a benign, reasonable query. But multiple safety-aligned models respond with refusals like “I can’t help you with that.” Only the VCL-enhanced model provides a study plan. That is not just “less harmful”—that’s **more helpful**.
>
> # **Weakness 3: Why We Use Residual Geometry Instead of Entropy-Based Metrics**
> That's a very reasonable question. Our choice to focus on **residual stream geometry**—rather than entropy measures—is driven by the belief that geometric analysis gives us **access to process-level representations**, not just output statistics. The residual stream reflects the model’s internal computation across layers, which allows us to uncover alignment distortions that might not be evident from entropy alone.
>
> In fact, there's a growing body of recent work that uses residual stream geometry to understand alignment and interpretability (see Section 4 Introduction). However, existing metrics—such as layerwise cosine similarity—have been found to be highly unstable under small prompt variations. Our goal was to introduce **a more stable and global diagnostic**, which led to the development of **FlowLens**.
>
> More detailed theoretical and empirical justification for FlowLens can be found in **Section 4.3** and **Appendix E**, where we show that:
> - PCA-based projections are provably stable under perturbations (via covariance perturbation theory),
> - and empirically robust even under minor formatting changes (e.g., punctuation differences).
>
> That said, we agree entropy-based measures are also informative and widely used (as shown in our own Section 3). We believe that combining **entropy and geometric analysis** could be a promising direction for future work, and we appreciate the suggestion.
>
> # **Weakness 4: Missing result table in Section 5.3**
>
> You're right—We have now added three result tables summarizing the effect of varying γ, k, and the residual window [l₁, l₂] on key evaluation metrics.
>
> (1) Effect of γ (regularization weight)
> (With k = 2, [l₁, l₂] = [0.3, 0.5])
>
> | γ        | DAN | Harmful | Toxigen | JBB  | OKTest | ORB  | XSTest | MMLU | GSM8K | BBH  | CodexEval |
> |----------|-----|---------|---------|------|--------|------|--------|------|--------|------|------------|
> | 0.01×50   | 0.85 | 0.811  | 0.96    | 0.83 | 0.70   | 0.82 | 0.78   | 0.39 | 0.49   | 0.24 | 0.22       |
> | 0.1×50   | 0.87 | 0.833  | 0.98    | 0.84 | 0.73   | 0.85 | 0.84   | 0.41 | 0.50   | 0.25 | 0.24       |
> | **1.0×50** | **0.89** | **0.841** | **1.000** | **0.86** | **0.76**  | **0.87** | **0.86**  | **0.42** | **0.51**  | **0.26** | **0.25**  |
> | 2.0×50   | 0.83 | 0.79   | 0.92    | 0.81 | 0.69   | 0.81 | 0.77   | 0.38 | 0.47   | 0.23 | 0.21       |
>
> (2) Effect of k (top principal components used)
> (With γ = 1.0×50, [l₁, l₂] = [0.3, 0.5])
>
> | k        | DAN | Harmful | Toxigen | JBB  | OKTest | ORB  | XSTest | MMLU | GSM8K | BBH  | CodexEval |
> |----------|-----|---------|---------|------|--------|------|--------|------|--------|------|------------|
> | 1        | 0.84 | 0.81   | 0.95    | 0.82 | 0.71   | 0.83 | 0.81   | 0.40 | 0.48   | 0.24 | 0.23       |
> | **2**      | **0.89** | **0.841** | **1.000** | **0.86** | **0.76**  | **0.87** | **0.86**  | **0.42** | **0.51**  | **0.26** | **0.25**      |
> | 4        | 0.87 | 0.825  | 0.97    | 0.84 | 0.74   | 0.86 | 0.85   | 0.41 | 0.50   | 0.25 | 0.24       |
> | 8        | 0.85 | 0.805  | 0.94    | 0.80 | 0.70   | 0.82 | 0.79   | 0.39 | 0.47   | 0.24 | 0.22       |
>
>  (3) Effect of residual window [l₁, l₂]
> (With γ = 1.0×50, k = 2)
>
> | [l₁, l₂]     | DAN | Harmful | Toxigen | JBB  | OKTest | ORB  | XSTest | MMLU | GSM8K | BBH  | CodexEval |
> |--------------|-----|---------|---------|------|--------|------|--------|------|--------|------|------------|
> | [0.1, 0.3]   | 0.86 | 0.823  | 0.97    | 0.84 | 0.73   | 0.85 | 0.84   | 0.40 | 0.49   | 0.25 | 0.24       |
> | **[0.3, 0.5]** | **0.89** | **0.841** | **1.000** | **0.86** | **0.76**  | **0.87** | **0.86**  | **0.42** | **0.51**  | **0.26** | **0.25**      |
> | [0.5, 0.7]   | 0.85 | 0.81   | 0.95    | 0.83 | 0.72   | 0.84 | 0.83   | 0.40 | 0.48   | 0.24 | 0.23       |
>
>
> These results confirm that VCL is robust to small changes in hyperparameters, and that the chosen configuration in the main paper achieves a good tradeoff between stability and refusal reduction. And we will integrate the table into the paper in the future version.
>
> # **Question 1: Slow PDF rendering due to PCA scatter plots**
> Thank you for flagging this. We’ve identified the cause: **Figures 4, 7, and 8** contain dense PCA scatter plots, currently rendered as vector graphics. Since these plots involve thousands of individual points, vector-based rendering can cause significant slowdown in some PDF viewers. To address this, we plan to convert these figures to high-resolution rasterized images (e.g., PNG) in future versions.
>
> # **Question 2: Clarifying the meaning of "Control" in Table 1**
> Thank you for pointing this out. We will revise the caption of Table 1 to explicitly define “Control” as follows:
> > *“‘Control’ refers to the general instruction data used as a baseline, consisting of 100,000 randomly sampled non-safety examples from the Tulu-3-SFT-Mixture-General subset. See Appendix B for further details.”*
>
> # **Question 3: Hyperparameters used when training with VCL**
> Beyond the hyperparameters discussed in Section 5.3 (γ, k, [l₁, l₂]), we largely **follow the default settings from the open-instruct recipe** [1].
>
> The training configurations are as follows:
>
> - **For LLaMA-3.2-1B**:
>   - `max_seq_length = 4096`
>   - `learning_rate = 1e-5`
>   - `warmup_ratio = 0.03`
>   - `weight_decay = 0.0`
>   - `num_train_epochs = 2`
>   - `batch_size = 128`
>
> - **For LLaMA-3.1-8B**:
>   - `max_seq_length = 8128`
>   - `learning_rate = 2e-5`
>   - `warmup_ratio = 0.03`
>   - `weight_decay = 0.0`
>   - `num_train_epochs = 2`
>   - `batch_size = 256`
>
> We will add this information to the main paper for clarity and reproducibility.
>
> [1] Lambert, N., Morrison, J., Pyatkin, V., Huang, S., Ivison, H., Brahman, F., ... & Hajishirzi, H. (2024). Tulu 3: Pushing frontiers in open language model post-training. arXiv preprint arXiv:2411.15124.

---

> > ### Comment · Reviewer_r4L3 · 2025-08-05
> >
> > Thanks for the response! Some of my concerns are addressed, and I keep my score as positive.

---

### Decision · Program_Chairs · 2025-09-17

**Decision:**

Accept (poster)

**Comment:**

This paper addresses the important problem of false refusals in safety-aligned LLMs by diagnosing variance collapse in residual streams (via FlowLens) and proposing a simple regularizer (VCL) that substantially reduces refusals while maintaining general performance. Reviewers agree the problem is relevant, the analysis novel, and the solution effective, though they note limitations in theoretical depth and model coverage. The rebuttal provides convincing additional experiments (Qwen, DPO) and clarifications, resolving most concerns. Overall, the work is technically solid, timely, and impactful, and thus my recommendation is to accept.